# Association between Serum Concentration of Carotenoid and Visceral Fat

**DOI:** 10.3390/nu13030912

**Published:** 2021-03-11

**Authors:** Mai Matsumoto, Hiroyuki Suganuma, Naoki Ozato, Sunao Shimizu, Mitsuhiro Katashima, Yoshihisa Katsuragi, Tatsuya Mikami, Ken Itoh, Shigeyuki Nakaji

**Affiliations:** 1Innovation Division, KAGOME CO. LTD., 17 Nishitomiyama, Nasushiobara, Tochigi 329-2762, Japan; Hiroyuki_Suganuma@kagome.co.jp (H.S.); Sunao_Shimizu@kagome.co.jp (S.S.); 2Department of Active Life Promotion Sciences, Graduate School of Medicine, Hirosaki University, 5 Zaifu-cho, Hirosaki, Aomori 036-8562, Japan; oozato.naoki@kao.com (N.O.); katashima.mitsuhiro@kao.com (M.K.); katsuragi.yoshihisa@kao.com (Y.K.); 3Health & Wellness Products Research Laboratories, Kao Corporation, Tokyo 131-8501, Japan; 4Department of Vegetable Life Science, Graduate School of Medicine, Hirosaki University, 5 Zaifu-cho, Hirosaki, Aomori 036-8562, Japan; itohk@hirosaki-u.ac.jp; 5Innovation Center for Health Promotion, Graduate School of Medicine, Hirosaki University, 5 Zaifu-cho, Hirosaki, Aomori 036-8562, Japan; tmika@hirosaki-u.ac.jp (T.M.); nakaji@hirosaki-u.ac.jp (S.N.); 6Department of Stress Response Science, Center for Advanced Medical Research, Graduate School of Medicine, Hirosaki University, 5 Zaifu-cho, Hirosaki, Aomori 036-8562, Japan; 7Department of Social Medicine, Graduate School of Medicine, Hirosaki University, 5 Zaifu-cho, Hirosaki, Aomori 036-8562, Japan

**Keywords:** carotenoid, visceral fat, metabolic syndrome, vegetable intake, resident-based cross-sectional study, healthy subjects

## Abstract

Consumption of fruits and vegetables rich in carotenoids has been widely reported to prevent cardiovascular diseases. However, the relationship between serum carotenoid concentrations and visceral fat area (VFA), which is considered a better predictor of cardiovascular diseases than the body-mass index (BMI) and waist circumference, remains unclear. Therefore, we examined the relationship in healthy individuals in their 20s or older, stratified by sex and age, to compare the relationship between serum carotenoid concentrations and VFA and BMI. The study was conducted on 805 people, the residents in Hirosaki city, Aomori prefecture, who underwent a health checkup. An inverse relationship between serum carotenoid concentrations and VFA and BMI was observed only in women. In addition, the results were independent of the intake of dietary fiber, which is mainly supplied from vegetables as well as carotenoids. This suggests that consumption of a diet rich in carotenoids (especially lutein and beta-carotene) is associated with lower VFA, which is a good predictor of cardiovascular disease, especially in women. This study is the first to comprehensively evaluate the association between serum carotenoid levels and VFA in healthy individuals.

## 1. Introduction

Carotenoids are red, orange, and yellow pigments widely distributed among plants, animals, and microorganisms. Because carotenoids have a strong ability to quench singlet oxygen, their ingestion is expected to prevent and improve various diseases in which reactive oxygen species (ROS) are involved in the onset or exacerbation of symptoms. Many studies have been conducted in this regard. Several epidemiological studies have shown that ingestion of carotenoids and carotenoid-rich vegetables and fruits leads to the amelioration of arteriosclerosis and hypertension and reduces the risk of chronic cardiovascular diseases, such as ischemic heart disease and cerebrovascular disease [1,2,3,4,5,6,7]. We also found that the higher the levels of carotenoids in the blood and skin, the healthier are many predictive markers of chronic cardiovascular disease in healthy individuals [8,9], suggesting that vegetable intake may be preventive against diseases in healthy as well as ill individuals. In addition to their role in reducing oxidative stress, some carotenoids have an inhibitory effect on the differentiation and hypertrophy of adipocytes [10], and some affect lipid and glucose metabolism [11,12]. Thus, copious intake of carotenoid-rich vegetables is also expected to inhibit metabolic syndrome (visceral fat syndrome), which is a crucial trigger of chronic cardiovascular disease and is thought to be important for its prevention [13].

Visceral fat area (VFA) is commonly measured by computed tomography (CT). However, because of the involvement of X-ray radiation, this measurement is not readily feasible. Therefore, the relationship between carotenoids and obesity was limited to the body-mass index (BMI) or waist circumference-based assessments. For example, in our previous study [8], high BMI was associated with low serum concentration of total carotenoid, whereas Suzuki et al. have reported an association between high waist circumference and low serum β-carotene levels in females [14]. Carotenoid concentrations in adipose tissue have also been measured, which have been shown to be related to waist circumference [15]. Sato et al. reported that there were misjudgments of more than 20% when they judged whether the VFA was equal to or larger than 100 cm^2^ using the criteria of waist circumference [16], so measuring VFA itself rather than waist circumference is considered preferable for subsequent disease prevention. However, as mentioned earlier, due to the difficulties in the measurement of VFA, the relationship between carotenoids and VFA has been evaluated only in a few previous studies [17]. The relationship between carotenoid intake, calculated from the food frequency questionnaire (FFQ), and VFA was evaluated in a few studies, wherein it was found that the higher the intake of β-carotene and lycopene, the lower the VFA in middle-aged and older men [17]. However, there has been no comprehensive assessment of the relationship between VFAs and levels of different carotenoids in the blood. To our knowledge, the age group for which the evaluations have been performed are also limited, and the studies evaluating a wide range of age groups, including younger age groups, are lacking.

Ryo et al. developed a noninvasive, easy-to-measure method for VFA, the results of which highly correlate with the measurements by CT [18]. They evaluated the relationship between VFA levels measured using the device developed by them and various biomarkers and environmental factors relevant to the field of medical examination and found a negative relationship between VFA and intake of dietary fiber (total and water-soluble), mainly taken through vegetables, as well as carotenoids [19]. However, the relationship between carotenoids and VFA has remained unclear.

In this study, we evaluated the relationship between serum carotenoid levels and VFA, including dietary habits, such as dietary fiber intake, in healthy individuals who participated in the Iwaki Health Promotion Project, a health examination project for a wide range of age groups.

## 2. Materials and Methods

### 2.1. Study Design and Subjects

This cross-sectional study was based on an annual health examination (the Iwaki Health Promotion Project) for the residents of the rural area of Hirosaki city, Aomori prefecture, Japan, conducted by Hirosaki University in May and June 2015. Of all participants (n = 1113), 805 (310 males and 495 females) were included as healthy subjects (Figure 1). Complete clinical data were available for these subjects, and they were not on any medication for dyslipidemia and had no history of serious diseases, such as cancer, stroke, cardiovascular diseases, liver diseases, kidney diseases, or diabetes. All procedures, including subject recruitments, were conducted in accordance with the Declaration of Helsinki and were approved by the ethics boards of Hirosaki University School of Medicine (2014-377, 2016-028) and KAGOME CO., LTD. (2015-R04). Written informed consent was provided by all subjects. This work has also been registered in the public information database UMIN-CTR by Hirosaki University and Kao Corporation (registration number: UMIN000030351).

### 2.2. Body Measurements

We measured VFA using the EW-FA90 (Panasonic Corporation, Osaka, Japan) based on an abdominal bioimpedance method developed at the Kao Corporation. BMI was calculated from the body weight and height, which were obtained by anthropometric measurements.

### 2.3. Self-Administered Questionnaire

On the day of the health examination, we collected the self-developed questionnaires that the participants had filled out in advance. The questionnaire contained information about sex, age, current smoking habits, exercise habits, medical history, and medications. We defined an exerciser as a person who habitually exercised at least 30 min a day and at least twice a week throughout the year. The volume of food consumed (g/day), including alcohol intake, was estimated using a brief-type self-administered diet history questionnaire (BDHQ) [20]. The daily intake for key food groups is calculated using BDHQ based on the past month’s dietary surveys.

### 2.4. Blood Sampling and Testing

Blood was collected from the median cubital vein after overnight fasting. The serum carotenoids were quantified at KAGOME CO., LTD. (Nagoya, Japan). Carotenoids (lutein, zeaxanthin, β-cryptoxanthin, α-carotene, β-carotene, and lycopene) were extracted and measured in accordance with the methods described previously [21,22]. We used a high-performance liquid chromatograph with a photodiode array detector (Prominence LC-30AD/Nexera X2 SPPD-M30A, SHIMADZU CORPORATION, Kyoto, Japan) for the measurements.

### 2.5. Statistical Analyses

As described in our previous report [8], the mean value of each measurement was stratified by sex (male and female) and compared using Student’s *t*-test. It was also stratified by age category (young, 20–39 years; middle-aged, 40–59 years; old ≥60 years) and compared using one-way ANOVA (post hoc: Bonferroni). Because previous studies have shown that carotenoid concentrations do not follow normality [8], Mann–Whitney *U* test and Kruskal–Wallis test (post hoc: Bonferroni) were used for the carotenoids.

Correlation analysis between serum concentrations of carotenoids and other factors (VFA, BMI, and foods) stratified by sex was performed using Pearson’s correlation coefficient.

Multiple regression analysis stratified by sex and age was conducted to evaluate the relationships between the serum concentrations of carotenoids and VFA or BMI. VFA and BMI were adopted as objective variables, whereas carotenoids were adopted as explanatory variables. Age, use of medication for some diseases (hypertension, hyperlipidemia, diabetes, dementia, rheumatism, and steroids), alcohol intake, current smoking habits, exercise habits, and total dietary fiber intake were adopted as adjustment factors according to the analysis. Analyses were performed using the R statistical package (version 3.5.1, R Foundation for Statistical Computing, Vienna, Austria) and EZR (version 1.4) [23], and a *p*-value < 0.05 was considered statistically significant.

## 3. Results

### 3.1. Characteristics of the Study Subjects

The mean values of all measurements stratified by sex and age categories are shown in Table 1. Both smoking habit and exercise habit rates were higher in males than in females in total.

VFA and BMI were significantly higher in males in total. In males, VFA in the young individuals was lower than that in the middle-aged and old individuals, and it increased by age in females. BMI was significantly higher in old individuals compared with that in young ones only in females.

Serum concentrations of carotenoids were higher in females except for zeaxanthin and lycopene. Concentrations of serum lycopene were significantly lower in the old in both males and females. For most of the other carotenoids, the serum levels were higher in middle-aged and old individuals than in the young ones.

Dietary fiber intake was higher in older age groups. The intake of carrots and pumpkin, and root vegetables was higher in females than in males. For both males and females, the intake of “radish/turnip” and “pickles” was higher in the old than in the other age groups.

### 3.2. Relationship between Serum Carotenoid Concentrations and VFA/BMI

Table 2 shows simple correlation coefficients between serum concentrations of carotenoids and VFA or BMI. The correlation coefficient between α-carotene and β-carotene was found to be high in previous studies [8], and the β-carotene concentration was used as a representative for the assessment. 

In males, neither VFA nor BMI showed significant correlations with either total carotenoids or individual carotenoids. In contrast, both VFA and BMI were negatively correlated with total carotenoids, lutein, β-carotene, and lycopene in females. BMI also had a negative correlation with zeaxanthin in females.

Multiple regression analysis was performed using VFA and BMI as an objective variable and serum total carotenoid concentration as an explanatory variable for the three age categories and in total for each sex. The standardized regression coefficients for each of these are shown in Table 3 and Table 4.

In males, a direct association between VFA and lycopene was found, but it was significant only in the old individuals after age stratification. On the contrary, in females, the inverse relationship between lycopene and VFA or BMI disappeared after adjusting for other factors by multiple regression analysis.

Multiple regression analysis also revealed a significant inverse relationship between VFA or BMI and β-cryptoxanthin in females. The inverse relationships between VFA or BMI and concentrations of total carotenoid, lutein, and β-carotene were found in all age groups, both after adjusting for other factors and after age stratification.

### 3.3. Relationship between Serum Concentration of Carotenoids and Food Consumed

In the Japanese population, vegetables are the main sources of carotenoids. Therefore, we examined the correlation between food items related to vegetables in the dietary survey and serum concentrations of carotenoids (Figure 2).

Total carotenoid concentrations directly correlated to certain vegetable groups (green leafy vegetables, carrots and pumpkins, root vegetables, and 100% juice), with correlation coefficients greater than 0.2 in males. In females, total carotenoid concentrations showed statistically significant direct correlations with most of the vegetable groups, including raw vegetables, but the overall correlation coefficients were less than 0.2. Serum zeaxanthin level showed little significant correlation with vegetables in both males and females. Lycopene showed a significant direct correlation with tomatoes, which are considered the main source of lycopene in both sexes. On the contrary, lycopene showed an inverse correlation with pickles.

The association between serum carotenoid levels and intake volume of various menus and food groups was examined. Table 5 and Table 6 show the top five food groups in terms of the positive and negative correlation coefficient.

In females, serum concentrations of total carotenoid, lutein, and β-carotene showed significant direct correlations with foods made from soy (“tofu” and fermented soybeans “natto”). Zeaxanthin showed a significant direct correlation with eggs in both the sexes; β-cryptoxanthin showed a significant direct correlation with fruits, including citrus fruits, in both males and females. Lycopene was directly correlated with tomato fruits and foods in which processed tomatoes are often used in cooking, such as hamburgers and pasta.

## 4. Discussion

In this study, we stratified healthy Japanese individuals over 20 years of age by sex and age and found multiple significant associations between serum carotenoid levels and VFA. The associations are discussed in the following sections, considering the results of a dietary survey conducted using BDHQ.

### 4.1. Relationship between Serum Carotenoid Levels and VFA

In the simple correlation analysis, only in females, total carotenoid concentrations in the serum were significantly and inversely associated with VFA. These relationships were kept in the lifestyle-adjusted multiple regression analysis or even age-stratified analysis. In our previous study [8], a significant negative relationship between total carotenoid levels in the serum and BMI was found only in females, which is consistent with the results of the present study. Sex differences in the serum concentrations of carotenoids may be one of the factors that contributed to this difference between males and females. In the present study, the mean total carotenoid concentration in females was significantly higher than that in males (male: 1.15 ± 0.53 μg/mL and female: 1.67 ± 0.75 μg/mL). If a certain threshold level of carotenoids was required to affect VFA, and many males do not meet the threshold, it may be possible that no significant relationship was found between carotenoid levels and VFA in males. It might also be possible that there are some differences in the carotenoid metabolism between the sexes. However, there is no concrete evidence for differences in carotenoid absorption or metabolism based on sex. In addition, differences in lifestyle between males and females may have influenced the results. Males are more likely than females to smoke, exercise, and consume alcohol. These lifestyles might be strongly associated with VFA in males. In a previous report [14], a relationship between waist circumference and carotenoids was found only in females, and the study speculated that smoking could be a contributing factor. In this study, we adjusted for these lifestyle factors in multiple regression analysis, but the effect of some of them on VFA is so large that it may not have been fully adjusted for males. Thus, although the causes of sex differences are considered diverse, there is no definitive evidence to support either of the hypotheses, and further research is needed. However, at least in females, higher serum total carotenoid levels may be beneficial for preventing cardiovascular disease, considering that VFA, a better predictor of cardiovascular disease than BMI, was also negatively associated with total carotenoid levels.

There are four hypotheses for the possible mechanisms by which carotenoids may seem to be useful in preventing cardiovascular disease. First, some physiological effects of carotenoids are hypothesized to be responsible. Carotenoids are known to have a variety of physiological effects [24]. A typical example is the ability to quench singlet oxygen, one of the ROS. ROS have been reported to be the cause of obesity [25], suggesting that reduction in their levels may inhibit obesity. One of the major factors for the increase in VFA is the accumulation of adipocytes. There are also reports that fucoxanthin, a carotenoid, suppresses the accumulation of adipocytes and, thereby, suppresses the accumulation of visceral fat [26]. In other words, carotenoids may directly inhibit the accumulation of adipocytes, which causes obesity and the accumulation of visceral fat. In addition, it has been reported that β-carotene concentrations are 50% lower in adipocytes of obese individuals compared to non-obese individuals [27], suggesting that carotenoids could be consumed under obese conditions to protect against obesity.

The second hypothesis is that it is because of the physiological action of vegetable components other than carotenoids. As previously reported, serum carotenoid levels are positively correlated with vegetable intake [8]. In addition, in the present study, serum levels of many carotenoids were positively associated with vegetable intake. In addition to vegetable intake, supplements are also assumed to be a source of carotenoids, but there was no data on the consumption of dietary supplements in this study. However, carotenoids supplements were reported to be taken by 2.4% of men and 4.4% of women in Japan [28], and we believe that the impact of those supplements on carotenoid intake was limited. Thus, individuals with high carotenoid levels may also have higher intakes of other vegetable ingredients. Dietary fiber is one of the components that is assumed to inhibit various chronic diseases. Dietary fiber has been previously suggested to reduce obesity and visceral fat accumulation [29]. However, the inverse associations between carotenoid levels and VFA were observed even though we added dietary fiber intake as an adjustment factor in the multiple regression analysis. Therefore, the associations are considered independent of the effect of dietary fiber. However, it cannot be denied that the influence of vegetable components other than dietary fiber may be indirectly observed.

Third, it is hypothesized that high vegetable intake leads to a reduction in energy intake. If this hypothesis is correct, it is thought that persons with a high intake of dietary vegetables and high serum carotenoid levels have a relatively low intake of nutrients, such as carbohydrates and lipids, and inhibit obesity and visceral fat accumulation. However, there was no significant correlation between total energy intake and total carotenoid concentration (male: r = 0.08, *p* = 0.172; female: r = 0.03, *p* = 0.56). Therefore, the inverse association between carotenoid concentrations and VFA observed in this study is unlikely to be due to a decrease in energy intake associated with high vegetable intake.

The fourth hypothesis is that it is caused by other healthy lifestyle aspects that are strongly associated with vegetable intake. Individuals with high health literacy and high vegetable intake may also have other healthy lifestyle habits, such as exercise habits and low breakfast absenteeism [30]. Thus, the inverse association observed in this study may be indirectly influenced by some healthy lifestyle habit that correlates with vegetable intake, rather than by direct effects of carotenoids or other vegetable components. However, the relationship between carotenoid levels and VFA is unlikely to be due to the presence or absence of smoking or exercise habits because exercise and smoking habits are included as adjustment factors in the multiple regression analysis. However, the influence of healthy eating habits, especially eating habits other than vegetable intake, cannot be denied. To clarify the dietary habits associated with high serum levels of these carotenoids, a simple correlation analysis was performed between various dietary groups calculated using BDHQ and serum carotenoid levels. The results suggested that the intake of soybean foods (natto and tofu) was also high in individuals with high serum levels of lutein and β-carotene. Several previous reports have shown that aggressive intake of soy food has an inhibitory effect on cardiovascular disease [31]. On the contrary, serum levels of lutein and β-carotene were inversely associated with the intake of hamburgers, fried foods, ramen, and beer. It is possible that diets high in vegetables and soybean foods and lifestyle habits that do not lead to excessive intake of carbohydrates and lipids could have inhibited the accumulation of VFA. However, more detailed studies evaluating the relationship between dietary intake and VFA are needed to confirm this.

To investigate the first hypothesis mentioned above, which proposes the direct effect of carotenoids, we analyzed the relationship between serum concentrations of individual carotenoids and VFA.

### 4.2. Relationship between Serum Concentrations of Lutein, β-Carotene, and β-Cryptoxanthin and VFA

In females, higher serum levels of lutein and β-carotene were associated with lower levels of VFA. This association and total carotenoid concentrations were maintained significantly in all age groups when stratified by age. The mean serum concentration of these two carotenoids is higher than that of the other carotenoids, and the sum of the two carotenoids is almost 60% of the total carotenoid concentration. These two carotenoids are ingested through the intake of many foods (mainly vegetables). For example, serum lutein levels were positively correlated not only with the intake of abundant green leafy vegetables [32] but also with that of raw vegetables, cooked cabbage, and pickles. β-Carotene was also positively correlated with the intake of all vegetable groups except raw vegetables and could be ingested through vegetables, in general.

The antioxidant effect common to carotenoids (including these two types of carotenoids [24]) was thought to have an inhibitory effect on the accumulation of VFA and obesity by inhibiting chronic inflammation. For example, β-carotene has been reported to be negatively associated with obesity and serum triglycerides [8]. It has been reported that β-carotene and its metabolites by β-carotene-9′,10′-dioxygenase control lipid metabolism within adipocytes, inhibit adipocyte maturation transformation, and suppress cell differentiation [33]. However, further studies are needed to elucidate the mechanisms by which lutein and β-carotene affect VFA.

Serum levels of β-cryptoxanthin were significantly negatively associated with VFA in females of all ages. The main source of β-cryptoxanthin was fruits, especially citrus fruits, similar to that in the Mikkabi cohort study conducted in Japan [34]. β-Cryptoxanthin has also been shown to be negatively associated with the markers of chronic cardiovascular disease [35]. Although the mechanisms by which β-cryptoxanthin prevents obesity have not been entirely clarified, it was suggested that β-cryptoxanthin might induce PPAR-α [36], antagonize PPAR-γ [37], and upregulate Keap1/Nrf2 pathway [38]. In the present study, lower BMI, but not VFA, was significantly associated with higher serum levels of β-cryptoxanthin in middle-aged females. The absence of this relationship with VFA could be due to the relatively low serum concentration of β-cryptoxanthin in citrus fruits (this study was conducted in May and June when the citrus fruits are not in season). Therefore, we speculate that a significant relationship between β-cryptoxanthin and VFA could have been observed if the studies would have been conducted during winter when the intake of citrus fruits is high. However, in-depth studies are needed to demonstrate the reduction in VFA by β-cryptoxanthin ingestion.

### 4.3. Relationship between Lycopene Levels and VFA

In males, a direct association was found between serum lycopene concentrations and VFA. The relationship was maintained even after stratification by age. However, lycopene has not been reported to increase VFA. Conversely, lycopene, as well as lutein and β-carotene, have potent antioxidant effects and may prevent obesity through the elimination of ROS, and there has been a report showing an inverse association between lycopene and VFA [17]. Takagi et al. showed that taking a beverage containing high lycopene reduced abdominal circumference in obese Japanese men by a small intervention study [39]. In the present analysis, a significant relationship was found, especially in the old male group, but the serum levels of lycopene were significantly lower in the old than in the other age groups. In addition, it has been reported that the low lycopene levels in Japanese individuals do not correlate with obesity [17]. Thus, the positive association between serum lycopene concentration and VFA is likely not a direct effect of lycopene. In addition, serum levels of lycopene were significantly positively correlated with the frequency of consumption of hamburgers and pastas. These foods are representative of tomato dishes, but they are also rich in carbohydrates and lipids. Previous reports [19] have also shown that monounsaturated fatty acids, which are abundant in olive oil, are positively correlated with VFA and may have been influenced by other nutrients in tomato dishes.

### 4.4. Study Limitations

This is a cross-sectional study and cannot be used to infer causality. Further studies are needed to determine whether the results can be extrapolated to other areas or other races, as the study included residents from some parts of Aomori prefecture. In addition, the BDHQ used in this study is a dietary frequency survey method, which has been validated [20], but it is possible that declaration errors occurred when compared to the weighing method, which is the golden standard for the dietary survey.

## 5. Conclusions

This is the first study to evaluate the association between serum carotenoids levels and VFA in healthy individuals. Although a significant association between high serum levels of carotenoids and low BMI, an indicator of obesity, was previously reported, we found that high serum levels of multiple carotenoids and total carotenoids were also associated with low VFA levels in females. These associations were independent of the intake of dietary fiber abundant in vegetables. Ingestion of carotenoid-rich vegetables (particularly lutein and β-carotene) may be associated with lower VFA, a good predictor of cardiovascular disease, especially in women.

## Figures and Tables

**Figure 1 nutrients-13-00912-f001:**
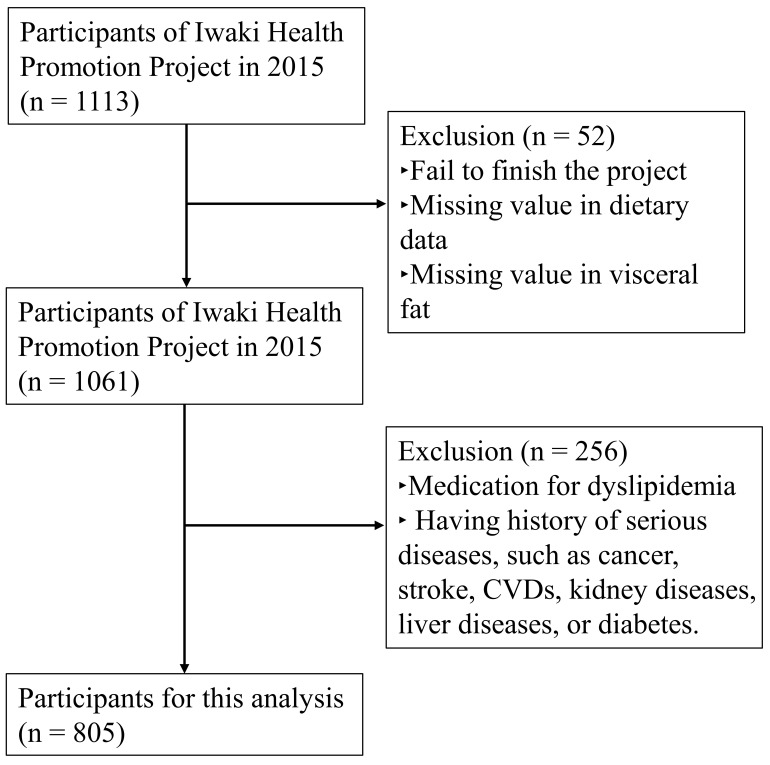
Flowchart for the recruitment of participants.

**Figure 2 nutrients-13-00912-f002:**
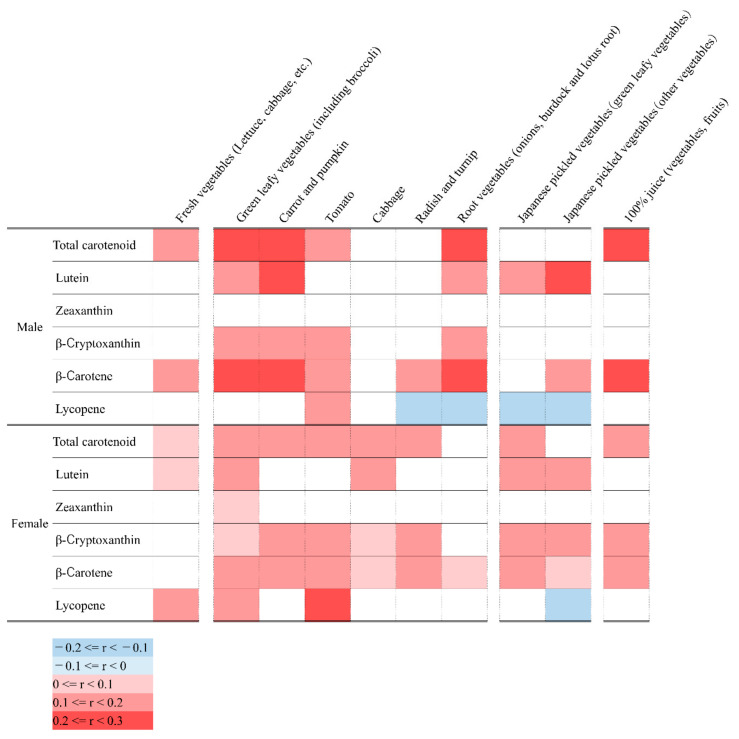
Correlation coefficients between serum carotenoids and consumed vegetables.

**Table 1 nutrients-13-00912-t001:** Characteristics of the study subjects.

Measurement Item	All	Male				Female			
		All	Young (20–39 Years)	Middle-Aged (40–59 Years)	Old (≥60 Years)	All	Young (20–39 Years)	Middle-Aged (40–59 Years)	Old (≥60 Years)
N	805	310	103	118	89	495	120	188	187
Age, year	51.1 ± 14.8	49.1 ± 14.8	32.5 ± 4.6	49.6 ± 5.5	67.7 ± 5.8	52.3 ± 14.6 **	32.5 ± 5.1	50 ± 5.8	67.3 ± 5.8
Alcohol intake, g/day	11 ± 19	21.8 ± 24.2	19.8 ± 25.6	25.8 ± 24.8	19 ± 20.8	4.2 ± 10.1 ***	4.3 ± 9.9	6.2 ± 12.4	2.3 ± 6.9 ^b^
Smoking, %	19.5	32.6	37.9	37.3	20.2	11.3	11.7	17.6	4.8
Habitual exercise, %	9.2	11.3	12.6	11.9	9	7.9	9.2	6.9	8
Medication, %	19.3	16.5	2.9	13.6	36	21	0.8	11.2	43.9
VFA, cm^2^	79.11 ± 42.28	102.81 ± 44.96	92.2 ± 47.62	108.1 ± 41.09 ^a^	108.07 ± 45.03 ^a^	64.27 ± 32.74 ***	53.13 ± 32.36	63.31 ± 31.13 ^a^	72.4 ± 32.49 ^ab^
BMI, kg/m^2^	22.58 ± 3.46	23.61 ± 3.23	23.14 ± 3.86	24.13 ± 2.87	23.47 ± 2.8	21.94 ± 3.45 ***	21.22 ± 3.95	21.76 ± 3.16	22.58 ± 3.28 ^a^
Total carotenoid, μg/mL	1.47 ± 0.72	1.15 ± 0.53	1.07 ± 0.5	1.13 ± 0.53	1.28 ± 0.56 ^a^	1.67 ± 0.75 ***	1.5 ± 0.68	1.61 ± 0.68	1.84 ± 0.84 ^ab^
Lutein, μg/mL	0.34 ± 0.14	0.3 ± 0.13	0.25 ± 0.1	0.31 ± 0.12 ^a^	0.35 ± 0.15 ^a^	0.36 ± 0.15 ***	0.3 ± 0.11	0.35 ± 0.15 ^a^	0.41 ± 0.16 ^ab^
Zeaxanthin, μg/mL	0.06 ± 0.02	0.06 ± 0.02	0.06 ± 0.02	0.06 ± 0.02	0.06 ± 0.02	0.06 ± 0.02	0.06 ± 0.02	0.06 ± 0.02	0.06 ± 0.02
β-Cryptoxanthin, μg/mL	0.15 ± 0.1	0.11 ± 0.06	0.1 ± 0.05	0.1 ± 0.05	0.14 ± 0.08 ^ab^	0.17 ± 0.11 ***	0.13 ± 0.06	0.17 ± 0.11 ^a^	0.21 ± 0.12 ^ab^
α-Carotene, μg/mL	0.17 ± 0.16	0.13 ± 0.14	0.13 ± 0.15	0.13 ± 0.13	0.14 ± 0.14	0.2 ± 0.16 ***	0.21 ± 0.19	0.19 ± 0.12	0.21 ± 0.17
β-Carotene, μg/mL	0.5 ± 0.42	0.3 ± 0.28	0.25 ± 0.25	0.28 ± 0.28	0.39 ± 0.28 ^ab^	0.62 ± 0.44 ***	0.5 ± 0.36	0.57 ± 0.36 ^a^	0.74 ± 0.52 ^ab^
Lycopene, μg/mL	0.25 ± 0.13	0.24 ± 0.13	0.28 ± 0.14	0.24 ± 0.12	0.19 ± 0.13 ^ab^	0.26 ± 0.14	0.3 ± 0.13	0.27 ± 0.13	0.21 ± 0.12 ^ab^
BDHQ									
Total dietary fiber, g/day	10.74 ± 4.40	11.11 ± 4.74	9.55 ± 3.52	10.58 ± 4.23	13.63 ± 5.56 ^ab^	10.51 ± 4.16	8.98 ± 3.75	10.36 ± 3.93 ^a^	11.63 ± 4.33 ^ab^
Soluble dietary fiber, g/day	2.66 ± 1.20	2.72 ± 1.32	2.33 ± 0.99	2.57 ± 1.18	3.38 ± 1.56 ^ab^	2.62 ± 1.13	2.26 ± 1.01	2.60 ± 1.08 ^a^	2.86 ± 1.19 ^a^
Insoluble dietary fiber, g/day	7.73 ± 3.05	8.04 ± 3.24	6.95 ± 2.50	7.67 ± 2.90	9.79 ± 3.72 ^ab^	7.54 ± 2.90 *	6.46 ± 2.61	7.42 ± 2.71 ^a^	8.37 ± 3.04 ^ab^
Fresh vegetables, g/day	25 ± 19.88	25.31 ± 20.77	23.94 ± 20.38	26.56 ± 20.33	25.24 ± 21.89	24.8 ± 19.32	23.26 ± 18.81	26.33 ± 19.04	24.25 ± 19.91
Green leafy vegetables, g/day	28.89 ± 28.61	26.73 ± 27.81	26.09 ± 24.93	25.28 ± 29.7	29.4 ± 28.5	30.25 ± 29.04	27.64 ± 25.94	31.95 ± 31.01	30.21 ± 28.89
Carrot and pumpkin, g/day	16.77 ± 15.07	14.23 ± 13.52	12.9 ± 12.13	13.84 ± 14.64	16.28 ± 13.39	18.36 ± 15.78 ***	17.08 ± 16.07	17.33 ± 14.01	20.22 ± 17.11
Tomato, g/day	19.39 ± 21.72	18.49 ± 19.96	17.35 ± 19.82	17.9 ± 19.31	0.58 ± 21.01	19.95 ± 22.76	17.14 ± 16.98	20.25 ± 23.81	21.45 ± 24.79
Cabbage, g/day	29.31 ± 26.34	30.27 ± 27.24	26.29 ± 22.66	30.84 ± 25.52	34.12 ± 33.34	28.71 ± 25.76	25.34 ± 23.09	27.5 ± 23.32	32.1 ± 29.23
Radish and turnip, g/day	15.31 ± 17.42	15.3 ± 18.84	12.86 ± 13.64	13.57 ± 17.25	20.4 ± 24.45 ^ab^	15.31 ± 16.49	13.12 ± 14.23	13.47 ± 14.28	18.57 ± 19.25 ^ab^
Root vegetables, g/day	29.88 ± 25.06	27.52 ± 26.27	24.74 ± 20.37	25.95 ± 24.16	32.83 ± 33.55	31.36 ± 24.18 *	30.29 ± 23.55	31.42 ± 24.36	31.99 ± 24.49
Pickled green leafy vegetables, g/day	6.08 ± 8.96	6.84 ± 9.05	4.59 ± 5.33	6.1 ± 7.5	10.42 ± 12.67 ^ab^	5.6 ± 8.88	3.36 ± 6.37	4.15 ± 6.81	8.5 ± 11.08 ^ab^
Pickled other vegetables, g/day	8.17 ± 10.84	8.46 ± 11.18	3.35 ± 4.66	9.39 ± 11.66 ^a^	13.13 ± 13.39 ^ab^	7.98 ± 10.62	4.27 ± 6.55	6.06 ± 8.32	12.3 ± 13.09 ^ab^
100% juice (vegetables and fruits), g/day	42.49 ± 74.38	39.83 ± 62.93	31.29 ± 50.86	40.6 ± 66.35	48.68 ± 69.92	44.16 ± 80.74	39.25 ± 58.06	41.69 ± 83.69	49.8 ± 89.8

The mean values ± SD of characteristics in the study subjects are shown. *, **, *** Mean values are significantly different from those in males (* *p* < 0.05, ** *p* < 0.01, *** *p* < 0.001). ^a^ Mean values are significantly different from those in the young group (*p* < 0.05). ^b^ Mean values are significantly different from those in the middle-aged group (*p* < 0.05). For carotenoids analysis, Mann–Whitney U test and Kruskal–Wallis test (post hoc: Bonferroni) were used. For the others, Student’s *t*-test and ANOVA (post hoc: Bonferroni) were used. VFA, visceral fat area; BMI, body mass index; BDHQ, brief-type self-administered diet history questionnaire.

**Table 2 nutrients-13-00912-t002:** Correlation coefficients between visceral fat area (VFA)/body-mass index (BMI) and carotenoids.

Carotenoids	Male		Female	
	VFA	BMI	VFA	BMI
Total carotenoid	0.003	0.050	−0.151 ***	−0.192 ***
Lutein	−0.017	−0.046	−0.095 *	−0.179 ***
Zeaxanthin	0.071	0.039	−0.046	−0.103 *
Β-Cryptoxanthin	−0.018	0.062	−0.027	−0.045
Β-Carotene	0.014	0.086	−0.138 **	−0.153 ***
Lycopene	0.070	0.062	−0.122 **	−0.145 **

* *p* < 0.05, ** *p* < 0.01, *** *p* < 0.001 (Pearson’s correlation coefficient).

**Table 3 nutrients-13-00912-t003:** Adjusted standard partial regression coefficient between visceral fat area (VFA) or body-mass index (BMI) and carotenoids (male).

Carotenoids	VFA				BMI			
	All	20–39	40–59	60–	All	20–39	40–59	60–
Total carotenoid	0.012	0.039	−0.119	0.121	0.052	0.094	0.017	0.104
Lutein	−0.059	−0.123	−0.026	−0.057	−0.063	−0.077	−0.060	0.000
Zeaxanthin	0.053	0.007	0.035	0.131	0.031	0.067	0.048	0.021
β-Cryptoxanthin	−0.012	−0.001	−0.036	0.058	0.073	0.114	0.093	0.101
β-Carotene	0.041	0.050	−0.068	0.155	0.117	0.147	0.059	0.198
Lycopene	0.123 *	0.189	−0.058	0.212 *	0.075	0.129	0.045	0.033

* *p* < 0.05. Adj: age, current smoking habits, exercise habits, alcohol intake, use of medications, and total dietary fiber intake.

**Table 4 nutrients-13-00912-t004:** Adjusted standard partial regression coefficient between visceral fat area (VFA) or body-mass index (BMI) and carotenoids (female).

Carotenoids	VFA				BMI			
	All	20–39	40–59	60–	All	20–39	40–59	60–
Total carotenoid	−0.210 ***	−0.256 **	−0.250 **	−0.180 *	−0.244 ***	−0.250 ***	−0.273 ***	−0.223 **
Lutein	−0.192 ***	−0.307 **	−0.168 *	−0.163 *	−0.258 ***	−0.323 ***	−0.234 **	−0.224 **
Zeaxanthin	−0.039	−0.147	−0.043	0.017	−0.010 *	−0.052	−0.135	−0.104
β-Cryptoxanthin	−0.110 *	−0.167	−0.144	−0.058	−0.113 *	−0.107	−0.162 *	−0.065
β-Carotene	−0.239 ***	−0.263 **	−0.253 **	−0.226 **	−0.237 ***	−0.231 *	−0.248 **	−0.220 **
Lycopene	−0.020	−0.008	−0.107	0.076	−0.078	−0.162	−0.109	−0.011

* *p* < 0.05, ** *p* < 0.01, *** *p* < 0.001. Adj: age, current smoking habits, exercise habits, alcohol intake, use of medications, and total dietary fiber intake.

**Table 5 nutrients-13-00912-t005:** Top five food groups that significantly correlated with serum carotenoids in males.

		Total Carotenoid	r	Lutein	r	Zeaxanthin	r	β-Cryptoxanthin	r	β-Carotene	r	Lycopene	r
Positive Correlation	1	Other fruits	0.27	Japanese pickled (other vegetables)	0.27	Egg	0.30	Fruits high in vitamin C	0.26	Other fruits	0.37	Ham	0.20
	2	Carrot and pumpkin	0.26	Seaweed	0.24	Pork and beef	0.15	Other fruits	0.26	Carrot and pumpkin	0.29	Pasta	0.19
	3	100% juice	0.25	“Natto”	0.21	Ham	0.13	Citrus	0.26	Root vegetables	0.27	Hamburger	0.16
	4	Egg	0.23	Carrot and pumpkin	0.20	Japanese liquor	0.13	Japanese sweets	0.18	Fruits high in vitamin C	0.26	Tomato	0.15
	5	Green leafy vegetables	0.23	Root vegetables	0.19	-		“Natto”	0.17	100% juice	0.25	Stir-fried meat	0.15
Negative Correlation	1	Beer	−0.25	Hamburger	−0.14	Fatty fish	−0.20	Beer	−0.35	Beer	−0.39	“Natto”	−0.20
	2	“Ramen”	−0.20	Coke	−0.14	Raw fish	−0.13	“Ramen”	−0.17	“Ramen”	−0.24	Japanese sake	−0.20
	3	Liver	−0.11	Ice cream	−0.13	Fried fish	−0.11	Japanese liquor	−0.14	Japanese liquor	−0.15	Japanese pickled (other vegetables)	−0.17
	4	-		-		-		-		Liver	−0.11	Fatty fish	−0.17
	5	-		-		-		-		-		Dried fish	−0.16

**Table 6 nutrients-13-00912-t006:** Top five food groups that significantly correlated with serum carotenoids in females.

		Total Carotenoid	r	Lutein	r	Zeaxanthin	r	β-Cryptoxanthin	r	β-Carotene	r	Lycopene	r
Positive Correlation	1	Other fruits	0.22	“Natto”	0.22	Egg	0.23	Citrus	0.30	Grilled fish	0.25	Tomato	0.25
	2	Fruits high in vitamin C	0.20	Grilled fish	0.19	Coffee	0.10	Other fruits	0.28	Other fruits	0.23	Hamburger	0.23
	3	“Tofu”	0.20	“Tofu”	0.18	Green leafy vegetables	0.10	Fruits high in vitamin C	0.22	“Tofu”	0.21	Fruits high in vitamin C	0.15
	4	Grilled fish	0.20	Green leafy vegetables	0.17	-		Grilled fish	0.21	Fruits high in vitamin C	0.21	Pasta	0.14
	5	100% juice	0.18	Fatty fish	0.16	-		Fatty fish	0.20	“Natto”	0.19	Fresh vegetables	0.12
Negative Correlation	1	Beer	−0.20	Hamburger	−0.18	Fried fish	−0.10	Beer	−0.23	Beer	−0.30	Japanese pickled (other vegetables)	−0.18
	2	“Ramen”	−0.15	Fried meat	−0.16	-		Ham	−0.11	“Ramen”	−0.18	Boiled fish	−0.15
	3	Fried meat	−0.12	Coke	−0.13	-		“Ramen”	−0.10	Japanese liquor	−0.14	Grilled fish	−0.15
	4	Japanese liquor	−0.10	Ham	−0.13	-		Fried food	−0.09	Fried meat	−0.13	Rice	−0.15
	5	Coke	−0.10	Fried meat	−0.12	-		-		Hamburger	−0.12	Dried fish	−0.14

## Data Availability

The data are not publicly available due to the ethical concerns. Data are available from the Hirosaki University COI Program Institutional Data Access/Ethics Committee for researchers who meet the criteria for access to the data.

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
