# Peer review of "Association between Serum Concentration of Carotenoid and Visceral Fat"

_nutrients, 2021, doi:10.3390/nu13030912_

Round 1
Reviewer 1 Report
Current manuscript entitled "Association between Serum Concentration of Carotenoid and Visceral Fat" is a very interesting and novel paper. Key finding is that high serum levels of multiple carotenoids and total carotenoids were also associated with low VFA levels in female subjects.
Introduction section: Please change the expression "Katashima of kao Corporation, one of the coauthors of this paper", it does not fit to a scientific paper.
Methods: This study does not have any info of how the necessary sample size was calculated.
Results: Figure no 2 is excellent, well done.
Discussion: well written, easy to follow
Author Response
Response to Reviewer 1’s Comments
Point: Current manuscript entitled "Association between Serum Concentration of Carotenoid and Visceral Fat" is a very interesting and novel paper. Key finding is that high serum levels of multiple carotenoids and total carotenoids were also associated with low VFA levels in female subjects.
Response: Dear Reviewer, Thank you for your valuable time and effort in reviewing our manuscript. We are grateful for your appreciation of the novelty of this study.
We have revised the entire manuscript following the below-mentioned insightful suggestion. Below we have provided point-by-point responses to each of your comments. We believe that our replies and the changes have addressed your raised concerns satisfactorily.
Point 1 Introduction section: Please change the expression "Katashima of kao Corporation, one of the coauthors of this paper", it does not fit to a scientific paper.
Response 1: We appreciate your comment. Accordingly, we have revised the text;
(Lines 86-87 in the revised version)
“Ryo et al. developed a noninvasive, easy-to-measure method for VFA, the results of which highly correlate with the measurements by CT [18].”
Point 2 Methods: This study does not have any info of how the necessary sample size was calculated.
Response 2: Apologies for the confusion. However, as this is an observational study, no particular sample size was set; therefore, no description was provided. Here, we have included all subjects after excluding those not fulfilling the inclusion criteria. We have described a flow chart describing the steps followed to recruit the patients in this study. However, to aid clarity, we have slightly modified the pre-existing text;
“Of all participants (n = 1113), 805 (310 males, 495 females) were included as healthy subjects (Figure 1). Complete clinical data were available for these subjects, and they were not on any medication for dyslipidemia and had no history of serious diseases, such as cancer, stroke, cardiovascular diseases, liver diseases, kidney diseases, or diabetes.” (Lines 103-107 in the revised version)
Point 3 Results: Figure no 2 is excellent, well done.
Response 3: Thank you for your appreciation.
Point 4 Discussion: well written, easy to follow
Response 4: Thank you for your appreciation.
Reviewer 2 Report
The manuscript is about the correlation between the serum carotenoid contend and visceral fat. The article is not really original as, for example in 2006 Suzuki et al, concluded as “abdominal fat accumulation is associated with oxidative stress as determined by low levels of serum carotenoids in females”.
Major revision
- In the abstract section the authors have said: “Therefore, we examined the relationship in healthy individuals……, to compare the relationship between serum carotenoids concentrations and BMI”. There are lot of references about this correlation, the authors should be focalized with the correlation between the visceral fat and the carotenoid content. Please reformulates the abstract and the manuscript.
- The authors have not commented the food supplement consumption, and this is a really important information. Please, could you inform about the food supplements, specially those which contain carotenoids? For example, the authors have said that: the results were independent of the intake of dietary fiber, which is mainly supplied from vegetables as well as carotenoids. And it could be due to the food supplement consumption.
Moreover, the authors could explain if the food supplements result in higher plasma carotenoid content, and apply considered statistical corrections.
- The introduction did not provide sufficient background and some relevant references are lacking. Please, update the bibliographic search, and modify the introduction and the discussion section.
Author Response
Response to Reviewer 2’s Comments
Point: The manuscript is about the correlation between the serum carotenoid contend and visceral fat. The article is not really original as, for example in 2006 Suzuki et al, concluded as “abdominal fat accumulation is associated with oxidative stress as determined by low levels of serum carotenoids in females”.
Response: Dear Reviewer, Thank you for your valuable time and effort in reviewing our manuscript. Following your insightful suggestions, we have revised the entire manuscript. Below, we have provided the point-by-point responses to each of your comments. We believe that our replies and the changes have addressed your raised concerns satisfactorily.
As you pointed out, the study was missing a discussion on waist circumference and carotenoids. However, the visceral fat area (VFA) is a good indicator of cardiovascular disease compared to waist circumference. As mentioned in the introduction, there are few reports on the VFA in the literature. Moreover, to the best of our knowledge, the correlation between VFA with blood carotenoid concentrations have not been reported in the past. Therefore, we believe that the topic of the present study is novel exploring that high serum levels of multiple carotenoids and total carotenoids were associated with low VFA levels in female subjects.
To aid clarity, we have slightly revised the Abstract and Discussion sections and added a new paragraph to the Introduction section of the revised manuscript. The revised texts are as follows;
In Abstract: “However, the relationship between serum carotenoid concentrations and visceral fat area (VFA), which is considered a better predictor of cardiovascular diseases than the body-mass index (BMI) and waist circumstance, remains unclear.” (Lines 24-27 in the revised version)
In Introduction: “ Therefore, the relationship between carotenoids and obesity was limited to the body-mass index (BMI) or waist circumference-based assessments. For example, in our previous study [8], high BMI was associated with low serum concentration of total carotenoid, whereas Suzuki et al. have reported an association between high waist circumference and low serum β-carotene levels in females [14]. Carotenoid concentrations in adipose tissue have also been measured, which have been shown to be related to waist circumference [15]. Sato et al. reported that there were misjudgments of more than 20% when they judged whether the VFA was equal to or larger than 100 cm2 using the criteria of waist circumference [16], so measuring VFA itself rather than waist circumference is considered preferable for subsequent disease prevention. However, as mentioned earlier, due to the difficulties in the measurement of VFA, the relationship between carotenoids and VFA has been evaluated only in a few previous studies [17].” (Lines 64-77 in the revised version)
In Discussion: “In a previous report [14], a relationship between waist circumference and carotenoids was found only in females, and the study speculated smoking could be a contributing factor.” (Lines 291-293 in the revised version)
Point 1: In the abstract section the authors have said: “Therefore, we examined the relationship in healthy individuals……, to compare the relationship between serum carotenoids concentrations and BMI”. There are lot of references about this correlation, the authors should be focalized with the correlation between the visceral fat and the carotenoid content. Please reformulates the abstract and the manuscript.
Response 1: We appreciate your thorough review and apologize for the mistake in the abstract. As suggested, we have changed the description of BMI in the discussion to focus on VFA. The revised text reads;
“ Therefore, we examined the relationship in healthy individuals in their 20s or older, stratified by sex and age, to compare the relationship between serum carotenoid concentrations and VFA and BMI.” (Lines 27-29 in the revised version)
“In the present study, lower BMI, but not VFA, was significantly associated with higher serum levels of β-cryptoxanthin in middle-aged females. The absence of this relationship with VFA could be due to the relatively low serum concentration of β-cryptoxanthin in citrus fruits (this study was conducted in May and June when the citrus fruits are not in season).” (Lines 388-392 in the revised version)
We removed the word “BMI” from lines 274, 276, 284, 286, 291, 294, 307, 326, 336, 344, 361, 363, 365, 382, 393, and 396 in the revised version.
Point 2: The authors have not commented the food supplement consumption, and this is a really important information. Please, could you inform about the food supplements, specially those which contain carotenoids? For example, the authors have said that: the results were independent of the intake of dietary fiber, which is mainly supplied from vegetables as well as carotenoids. And it could be due to the food supplement consumption.
Moreover, the authors could explain if the food supplements result in higher plasma carotenoid content, and apply considered statistical corrections.
Response 2: Thank you for the insightful comment. Unfortunately, we could not analyze the data on supplements because we did not obtain any information on them in this study. In Japan, carotenoid supplements were reported to be taken by 2.4% men and 4.4% women, and we believe that the impact of supplements is limited. However, we agree with your opinion that its influence cannot be ruled out on the results of this study; therefore, we revised the text as follows;
“In addition to vegetable intake, supplements are also assumed to be a source of carotenoids, but there was no data on the consumption of dietary supplements in this study. However, carotenoids supplements were reported to be taken by 2.4% of men and 4.4% of women in Japan [28], and we believe that the impact of those supplements on carotenoid intake was limited.” (Lines 317-322 in the revised version)
Point 3: The introduction did not provide sufficient background and some relevant references are lacking. Please, update the bibliographic search, and modify the introduction and the discussion section.
Response 3: Thank you for your useful comments, which helped us improve the quality of the paper. Following your suggestion, we have reviewed the existing information and revised the introduction and discussion section. We believe the improvement has adequately addressed the raised aspect; however, we would appreciate making further changes, if required. We revised the text as follows;
In Introduction: “Therefore, the relationship between carotenoids and obesity was limited to the body-mass index (BMI) or waist circumference-based assessments. For example, in our previous study [8], high BMI was associated with low serum concentration of total carotenoid, whereas Suzuki et al. have reported an association between high waist circumference and low serum β-carotene levels in females [14]. Carotenoid concentrations in adipose tissue have also been measured, which have been shown to be related to waist circumference [15]. Sato et al. reported that there were misjudgments of more than 20% when they judged whether the VFA was equal to or larger than 100 cm2 using the criteria of waist circumference [16], so measuring VFA itself rather than waist circumference is considered preferable for subsequent disease prevention. However, as mentioned earlier, due to the difficulties in the measurement of VFA, the relationship between carotenoids and VFA has been evaluated only in a few previous studies [17]. ” (Lines 64-77 in the revised version)
In Discussion: “In addition, it has been reported that β-carotene concentrations are 50% lower in adipocytes of obese individuals compared to non-obese individuals [27], suggesting that carotenoids could be consumed under obese conditions to protect against obesity.” (Lines 310-313 in the revised version)
“It has been reported that β-carotene and its metabolites by β-carotene-9’,10’-dioxygenase control lipid metabolism within adipocytes, inhibit adipocyte maturation transformation, and suppress cell differentiation [33]..” (Lines 377-389 in the revised version)
“Takagi et al. showed that taking a beverage containing high lycopene reduced abdominal circumference in obese Japanese men by a small intervention study [39]. ” (Lines 403-404 in the revised version)
Reviewer 3 Report
In the manuscript titled “Association between Serum Concentration of Carotenoid and Visceral Fat”, Matsumoto and colleagues evaluated the association between fruits and vegetables rich in carotenoids consumption and lower visceral fat and BMI.
The authors found gender differences in the association between carotenoid plasma levels and visceral fat accumulation. The results are very interesting and the analysis of this cohort of healthy individuals can be very useful to discriminate the effects of carotenoid-rich diet from other lifestyle factors that can potentially affect obesity and cardiovascular risk. The differences between males and females are very interesting and had to be further explored in the future. This reviewer understands that this is beyond the scope of this paper, and the authors explained very well the limitations of this study.
This reviewer has only few minor comments
- Statystical analysis: please add a refernce in the first paràgraf after the sentence “..carotenoid concentrations do not follow normality”
- Table 1: the legend is very confusing, its difficult to find the meaning of the symbols used. Maybe the authors could find a better way to explain the statistics.
Author Response
Response to Reviewer 3’s Comments
Point : In the manuscript titled “Association between Serum Concentration of Carotenoid and Visceral Fat”, Matsumoto and colleagues evaluated the association between fruits and vegetables rich in carotenoids consumption and lower visceral fat and BMI.
The authors found gender differences in the association between carotenoid plasma levels and visceral fat accumulation. The results are very interesting and the analysis of this cohort of healthy individuals can be very useful to discriminate the effects of carotenoid-rich diet from other lifestyle factors that can potentially affect obesity and cardiovascular risk. The differences between males and females are very interesting and had to be further explored in the future. This reviewer understands that this is beyond the scope of this paper, and the authors explained very well the limitations of this study.
Response: Dear Reviewer, Thank you for your valuable time and effort in reviewing our manuscript. We are grateful for your appreciation of the findings of this study.
We have revised the entire manuscript following your insightful suggestions and provided point-by-point responses to each of your comments. We believe that our replies and the changes have addressed your raised concerns satisfactorily.
Point 1: Statystical analysis: please add a refernce in the first paràgraf after the sentence “..carotenoid concentrations do not follow normality”
Response 1: We appreciate your comment. We apologize for the missing references. We have added to the text;
“Because previous studies have shown that carotenoid concentrations do not follow normality [8], Mann–Whitney U test and Kruskal–Wallis test [post hoc: Bonferroni] were used for the carotenoids.” (Lines 147-149 in the revised version)
Point 2: Table 1: the legend is very confusing, its difficult to find the meaning of the symbols used. Maybe the authors could find a better way to explain the statistics.
Response 2: Apologies for the confusion, and thank you for pointing this out. Accordingly, we have revised the Table footnote as follows;
“The mean values ± SD of characteristics in the study subjects are shown.
*, **, *** Mean values are significantly different from those in males (* p < 0.05, ** p < 0.01, *** p < 0.001).
aMean values are significantly different from those in the young group (p < 0.05).
bMean values are significantly different from those in the middle-aged group (p < 0.05).
For Carotenoids analysis, Mann–Whitney U test and Kruskal–Wallis test [post hoc: Bonferroni] were used. For the others, Student's t-test and ANOVA [post hoc: Bonferroni] were used.
VFA, visceral fat area; BMI, body mass index; BDHQ, brief-type self-administered diet history questionnaire.”
Round 2
Reviewer 2 Report
The authors have performed all the required changes, however the lack of the food supplements information is an important lacking point, which reduce the quality of the manuscript.